# Non-Invasive Method to Predict the Composition of Requeijão Cremoso Directly in Commercial Packages Using Time Domain NMR Relaxometry and Chemometrics

**DOI:** 10.3390/molecules27144434

**Published:** 2022-07-11

**Authors:** G. de Oliveira Machado, Gustavo Galastri Teixeira, Rodrigo Henrique dos Santos Garcia, Tiago Bueno Moraes, Evandro Bona, Poliana M. Santos, Luiz Alberto Colnago

**Affiliations:** 1Instituto de Química de São Carlos, Universidade de São Paulo, CP 369, São Carlos 13660-970, SP, Brazil; guilherme.oliveira.machado@usp.br (G.d.O.M.); rodrigogarciaquimico@yahoo.com.br (R.H.d.S.G.); 2Department of Microbiology, Institute of Biomedical Science, Universidade Tecnológica Federal do Paraná, Rua Deputado Heitor de Alencar Furtado, Curitiba 81280-340, PR, Brazil; gugalastri@hotmail.com; 3Depto. Engenharia de Biossistemas, Universidade de São Paulo, Av. Páduas Dias, Piracicaba 13418-900, SP, Brazil; tiagobuemoraes@gmail.com; 4Programa de Pós-Graduação em Tecnologia de Alimentos (PPGTA), Universidade Tecnológica Federal do Paraná, Rua Rosalina Maria Ferreira, Campo Mourão 87301-899, PR, Brazil; ebona@utfpr.edu.br; 5Embrapa Instrumentação, Rua XV de Novembro, São Carlos 13560-970, SP, Brazil

**Keywords:** TD-NMR, requeijão cremoso, CPMG, CWFP-T_1_, PLS, OPS

## Abstract

Low Field Time-Domain Nuclear Magnetic Resonance (TD-NMR) relaxometry was used to determine moisture, fat, and defatted dry matter contents in “requeijão cremoso” (RC) processed cheese directly in commercial packaged (plastic cups or tubes with approximately 200 g). Forty-five samples of commercial RC types (traditional, light, lactose-free, vegan, and fiber) were analyzed using longitudinal (T_1_) and transverse (T_2_) relaxation measurements in a wide bore Halbach magnet (0.23 T) with a 100 mm probe. The T_1_ and T_2_ analyses were performed using CWFP-T_1_ (Continuous Wave Free Precession) and CPMG (Carr-Purcell-Meiboom-Gill) single shot pulses. The scores of the principal component analysis (PCA) of CWFP-T_1_ and CPMG signals did not show clustering related to the RC types. Optimization by variable selection was carried out with ordered predictors selection (OPS), providing simpler and predictive partial least squares (PLS) calibration models. The best results were obtained with CWFP-T_1_ data, with root-mean-square errors of prediction (RMSEP) of 1.38, 4.71, 3.28, and 3.00% for defatted dry mass, fat in the dry and wet matter, and moisture, respectively. Therefore, CWFP-T_1_ data modeled with chemometrics can be a fast method to monitor the quality of RC directly in commercial packages.

## 1. Introduction

Low field time domain NMR (TD-NMR) relaxometry has great importance in food analyses due to its advantages, such as practicality, analytical productivity, rapid analysis, cost-effectiveness, and low-cost instrumentation [1,2,3,4]. Furthermore, since the applied radiofrequency (rf) is not attenuated by non-metallic materials (glass and plastic), TD-NMR analyses can be performed directly in the original packaging without requiring sample preparation nor generating waste [5,6].

The application of TD-NMR for food quality control in dairy products is one of its most widespread uses [7,8,9]. Some applications include detection and quantification of milk adulteration [10,11], determination of fat content in commercial products of milk powder [12], simultaneous quantification of fat and water content in cheese [13], characterization of yogurts [14], ice cream [15] and “requeijão cremoso” [16,17]. Requeijão cremoso (RC) is a type of processed cheese widely produced and consumed in Brazil [16,17]. It is prepared from the coagulation of pasteurized milk, with or without the addition of lactic cultures, followed by the addition of cream, water, and melting salt (a mixture containing trisodium citrate) [17]. Studies investigated water mobility in the homemade RC with the addition of galactooligosaccharide (GOS) using TD-NMR relaxometry and showed an increase in the transversal relaxation time (T_2_) with an increase of GOS [16]. Homemade RC with the addition of xylooligosaccharide (XOS), sodium reduction, and flavor enhancers (arginine and yeast extract) was also studied by TD-NMR relaxometry presented no differences in T_2_ in the samples with 50% reduced salt content and with the addition of 1% of arginine or yeast extract [17]. However, a reduction in the T_2_ values was observed in RC samples with lower fat content, which was associated with the decrease in mobility of water that strongly binds to the proteins. The addition of 3.3% of XOS also led to a substantial reduction in both transverse (T_2_) and longitudinal (T_1_) relaxation times.

The analysis of TD-NMR relaxometry of large samples, such as fresh fruits and oilseeds [18,19,20] and industrialized packaged food, can be performed using wide bore Halback, wide gap C and H types [21,22], or unilateral magnets [23,24]. The main advantage of Halback, C, or H types over the unilateral ones is the higher homogeneity of the magnetic field, allowing the analysis of the entire or large part of the sample. Conversely, unilateral magnets are cheaper and portable, and the magnet dimension does not limit the sample size. However, unilateral relaxometry has a strong magnetic field gradient, and only a small part of the sample is analyzed. Thus, the signal-to-noise ratio (SNR) in this equipment is very low and, consequently, the measuring time is much longer when compared to the developed in homogeneous magnets [21].

Most TD-NMR applications use the T_2_ measurements [1,25]. These analyses are performed using Carr-Purcell-Meiboom-Gill (CPMG) pulse sequence, a single shot sequence that is very insensitive to the flip angle error and rf magnetic field inhomogeneity [21]. Applications based on the longitudinal (T_1_) relaxation time are rarely used because the standard pulse sequences (inversion-recovery, saturation-recovery, progressive saturation, etc.) are not single shot sequences. Therefore, the measuring time is more than one order of magnitude longer than the CPMG measurement [21,26].

Another pulse sequence explored in the TD-NMR analysis is the continuous wave free precession (CWFP) [25,27,28], which is a special case of the steady-state free precession (SSFP) regime. In this sequence, rf pulses are separated by an interval shorter than the effective transverse relaxation time (T_2_*). When 90° pulses are used, CWFP signals depend on both T_1_ and T_2_ relaxation time. The first application of CWFP pulse sequence in low-field TD-NMR was proposed to enhance SNR to determine the hydrogen content in solvents and oil in seeds [21,25,27]. More recently, modifications in the CWFP pulse sequence were made, allowing to determine T_1_ time in a fast and accurate way. The sequence was named CWFP-T_1_ [27] and has been used in the analysis of packaged beef [29,30] and chicken breasts [31].

In this study, we demonstrated the viability of using CPMG and CWFP-T_1_ to predict moisture, fat, and defatted dry mass content in RC commercial samples. To the best of our knowledge, it is the first time that a method based on TD-NMR has been proposed for RC quality control. Moreover, no paper was found in the literature reporting the use of TD-NMR to predict moisture, fat, and defatted dry mass content in dairy food samples directly in packaged commercial products. The analysis was performed using a TD-NMR spectrometer based on a wide bore Halback magnet and the data were analyzed using the partial least squares (PLS) regression method. This strategy represents a fast and promising analytical method for the dairy food chain.

## 2. Materials and Methods

### 2.1. Samples

This study comprised 45 commercial RC samples (27 traditional, 9 light, 4 vegan, 3 lactose-free, and 2 fibers) in plastic cups or tube packages, containing approximately 200 g of RC. The samples were obtained from a local market (São Carlos, SP, Brazil) and stored at 5 °C.

### 2.2. TD-NMR Measurements

The RC TD-NMR analyses were evaluated in a non-invasive and non-destructive way directly in the original packaging. ^1^H TD-NMR experiments were evaluated using an SLK-IF-1399 NMR spectrometer (Spinlock Magnetic Resonance Solution, Cordoba, Argentine) equipped with a Halbach permanent magnet of 0.23 T (9 MHz for ^1^H) and a 100 mm probe. Transverse relaxation time (T_2_) measurements were performed using CPMG (Carr-Purcell-Meiboom-Gill) pulse sequence with a π/2 pulse width of 33 μs and π pulse width of 60 μs, echo time (τ) of 500 μs, 1000 echoes, and a recycle delay of 5 s. Values of longitudinal relaxation time (T_1_) were determined using CWFP-T_1_ (Continuous Wave Free Precession) pulse sequence [27] with a π/2 pulse width of 33 μs and π/10 pulse width of 4 μs, τ of 300 μs, 3000 echoes, and a recycle delay of 1 s. In both sequences, signals were averaged using 16 scans.

### 2.3. Chemical Analysis

The moisture content (grams of water per 100 g of RC product) was determined by drying the sample to a constant weight in an oven at 105 °C. Fat was determined gravimetrically after Soxhlet extraction of the dry matter (DM) using petroleum ether as extracting solvent. The extraction residue was considered the defatted dry mass (DDM). The fat content percentage was calculated in relation to dry (FDM) and wet matter (FWM), respectively.

### 2.4. Multivariate Analysis

The multivariate data analysis was performed using MATLAB R2021a (The Mathworks Inc., Natick, MA, USA) and PLS_Toolbox v.8.8 (Eigenvector Research Inc., Wenatchee, WA, USA).

The Principal Component Analysis (PCA) was used in a preliminary assessment to observe similarities and/or differences between the different types of RC samples. Before the analysis, the chemical data were auto-scaled and the TD-NMR decays (CPMG and CWFP-T_1_) were mean-centered.

The multivariate calibration models were developed based on partial least squares (PLS) regression to quantify moisture, FDM, FWM, and DDM in RC samples. The Kennard–Stone algorithm divided the data into calibration (75% of the samples) and validation (25% of the samples) data sets. Before the analysis, the TD-NMR decays (CPMG and CWFP-T_1_) were mean-centered. To improve the model reliability and develop simpler and predictive models, the algorithm ordered predictor selection (OPS) was used for variable selection. OPS Matlab routine can be freely obtained at http://www.deq.ufv.br/chemometrics [32].

The predictive ability of the final models was evaluated in terms of coefficient of determination (R^2^), root mean square error of prediction (RMSEP), the relative error of prediction (REP), and ratio performance to deviation (RPD).

## 3. Results and Discussion

### 3.1. TD-NMR Decays

Figure 1 shows the TD-NMR signals of three RC samples with different chemical compositions acquired using CPMG (Figure 1a) and CWFP-T_1_ (Figure 1b) pulse sequences. The TD-NMR relaxation curves are colored according to moisture content (black line = lowest, blue = medium, and red = highest moisture content). It is observed that the RC sample with the highest moisture (red line) had a significantly higher T_2_ value than the sample with the lowest moisture content (black line). This is in agreement with results reported in the literature [33,34], which showed that an increase in moisture resulted in a higher T_2_ relaxation time. Similar behavior was observed for T_1_ values; The sample with the highest moisture (red line) had a significantly higher T_1_ value than the sample with the lowest moisture (black line). T_2_ and T_1_ relaxation spectra of the three RC samples estimated by the Inverse Laplace transform (ILT) are shown respectively in Figure 1c,d [35]. The mean T_2_ and T_1_ relaxation curves show the presence of two populations, agreeing with previous studies in the literature, which reported that the T_2_ relaxation in RC is mainly bi-exponential [16,17].

The transverse relaxation times of the CPMG curves (Figure 1a) showed that the samples with low, medium, and high moisture could be resolved into a fast-relaxing time constant (T_21_) equals to 2, 3, and 12 ms, respectively, and a slow-relaxing time constant (T_22_) equals to 30, 41, and 78 ms, respectively. For CWFP-T_1_ signals, the RC samples with low, medium, and high moisture showed T_11_ values of 23, 43, and 58 ms, respectively, and a T_12_ of 329, 374, and 284 ms, respectively.

As RC contains between 55 and 80% of water and less than 22% of fat, the relaxation signals are dominated by the water relaxation time. As T_1_ > T_2_ is an indication that water mobility is restricted by the interaction of dry matter products, mainly lipids, proteins, and saccharides that correspond to 20–45% of the RC.

### 3.2. Exploratory Analysis

Preliminary data exploration with the PCA was performed in the chemical results (moisture, FDM, FWM, and DDM) to investigate natural differences and patterns among samples and highlight relationships between variables and classes. For this proposed, 45 commercially available RC samples composed of 27 traditional, 9 light, 4 vegan, 3 lactose-free, and 2 fibers, were disposed of in a matrix X (45 × 4). The results are presented on the PCA biplot (Figure 2). The first two principal components (PC1 and PC2) explained 99.8% of the total variance (81.2% and 18.6%, respectively) and provided a slight separation of traditional and vegan samples from the other RC types (light, lactose-free, and fiber). The examination of loadings revealed that the variables that most contributed to this separation were FWM and FDM, which are positioned positively on PC1. These results are in agreement with the chemical values, which revealed that the traditional and vegan samples had the highest fat content in dry (61.4 ± 9.3% and 67.4 ± 3.1%, respectively) and wet (23.2 ± 4.6% and 28.1 ± 1.7%, respectively) matter. Conversely, moisture and defatted dry mass are directly correlated with light, fiber, and lactose-free samples, since they are positioned negatively on PC1. The chemical results revealed that these samples showed the highest moisture values, ranging from 60.6 to 77.4%, and defatted dry mass, ranging from 11.1 to 20.1%.

CPMG and CWFP-T_1_ decays of the 45 samples were also analyzed by the PCA. Inspection of the PCA subspaces (bi- or three-dimensional score plot) did not reveal groups of samples with respect to the RC type (data not shown). These results suggest no significant differences in T_2_ and T_1_ time decay profiles were observed among the samples. However, due to the limited number of commercial samples, it was not possible to confirm that the method cannot be applied to distinguish the RC samples.

### 3.3. Regression Models

Initial PLS models were built by correlating the full TD-NMR decay and reference values of moisture, FDM, FWM, and DDM. As already mentioned (Section 2.4), the whole data set (27 traditional, 9 light, 4 vegan, 3 lactose-free, and 2 fibers) was divided into 33 samples for the training set and 12 samples for the test set. The models were optimized through the detection of outliers based on samples with extreme leverages, large residuals in the X block (spectral outliers) or in the Y block (prediction outliers) were detected at the 95% confidence level. The number of latent variables (LV) was chosen based on the smallest RMSECV values estimated using random subsets (10 data splits and 20 iterations). The figures of merit of the models showed that the predictions were not very accurate and thus could not be considered satisfactory. To confirm the model significance, a permutation test was performed, in which 100 interactions were performed comparing permuted with unpermuted models considering self-prediction and cross-validation. The Wilcoxon signed-rank test (Wilcoxon) indicates that the models are significant at a 95% confidence level. Moreover, in the self-prediction approach, the signed-rank test (sign test) and randomization *t*-test (Rand t test) were above 0.05, indicating that these models (permuted and nonpermutes) are not significantly different at the 95% limit. Since this outcome could be attributed to the possible presence of a high amount of irrelevant and/or noisy variables in the data set, the OPS variable selection was conducted.

The comparison of the PLS models with and without variable reduction for DDM quantification is shown in Table 1. The best models were obtained autoOPS algorithm, with a window of 10 and increments of 5, which is coherent with the literature [32]. The number of variables used to build the models was significantly reduced from 993 to 205 for CPMG data and from 5965 to 75 for CWFP-T_1_ data [36]. The models were obtained with 3 LV, providing parsimonious models. It can be seen that the variable selection steps increase the accuracy of the CWFP-T_1_ model. Comparing the results obtained with “full data”, the OPS-PLS model attained the best results with an RMSEP value of 1.38% and R^2^_pred_ of 0.90. Conversely, results suggested that the variable selection did not significantly improve the CPMG model in terms of RMSEP, RPD, and REP. A permutation test indicated that both OPS-PLS models were substantially different from the unpermuted ones and did not display overfitting at a 95% level.

The RPD value can also confirm the good prediction capacity of the CWFP-T_1_ model. RPD is a dimensionless figure of merit specifically used to evaluate the trueness of multivariate calibration models in absolute terms. According to the literature [37,38,39], desirable calibration models must have an RPD higher than 2.4, while RPD values between 2.4 and 1.5 are considered acceptable. Models with RPD less than 1.5 are considered unusable. Thus, the RPD value for the CWFP-T_1_ model was considered satisfactory, whereas the RPD value for the CPMG model was considered unusable. The REP was another figure of merit calculated to evaluate the trueness of the models, whose values were 9.55 and 8.70% w w^−1^, for the CPMG and CWFP-T_1_ models, respectively. This scenario is completely acceptable once the analysis was performed directly on the packaged commercial products. These results can also be graphically visualized in Figure 3, where the measured vs. predicted plots for both the calibration and validation sets are displayed.

Similar approaches were explored to quantify FWM, FDM, and moisture in the RC samples. The commercial samples presented FWM contents from 33 to 73% w w^−1^, FDM contents from 9 to 30% w w^−1,^ and moisture contents from 56 to 77% w w^−1^. The PLS models were developed using the autoOPS algorithm, with a window of 10 and increments of 5. The number of variables used to build the models was 205 and 75 for CPMG and CWFP-T_1_, respectively. The best models were obtained with 4 LV. Results showed that the CPMG models are inadequate to predict fat in wet and dry matter and moisture in the RC samples, with lower values of R^2^_pred_ (<0.28) and RPD_pred_ (<1.26) and higher values of RMSEP and REP (>7.71 w w^−1^) (Table 2). Furthermore, a permutation test indicated that models were not significantly different from the unpermuted ones at the 95% level. Conversely, the models developed using CWFP-T_1_ data showed to be suitable with R^2^_pred_ of 0.92, RMSEP of 4.71% w w^−1^, REP of 8.79% w w^−1^ and RPD_pred_ of 2.92 for fat in wet matter, R^2^_pred_ of 0.85, RMSEP of 3.28% w w^−1^, RPD_pred_ of 16.68% w w^−1^ and RPD_pred_ of 2.32 for fat in wet matter the model and R^2^_pred_ of 0.70, RMSEP of 3.00% w w^−1^, REP of 4.65% w w^−1^ and RPD_pred_ of 1.77 for moisture.

The reference vs. predicted values for (a) FWM (% w w^−1^), (b) FDM (% w w^−1^) and (c) moisture content (% w w^−1^) by using CWFP-T_1_ decays combined with the OPS selection variable method are shown in Figure 4.

Regression models were also evaluated by correlating the T_1_ or T_2_ values against the moisture, fat, and defatted dry matter contents (univariate approaches) and combining T_1_ and T_2_ values (multivariate approaches). However, results obtained with the full TD-NMR decays showed better predictability, with a higher correlation coefficient. Therefore, only these results were shown in detail in the manuscript. 

Methods based on near- and mid-infrared spectroscopy have been reported in the literature for moisture and fat determination in dairy products, especially in cheese [40]. Nevertheless, no study on RC samples was found. A procedure to quantify fat and moisture in cow mozzarella cheeses by using reflectance near-infrared spectroscopy has been developed and validated [37]. The PLS models were constructed within the ranges from 38.7 to 58.0% w w^−1^ on dry basis for fat and from 41.5 to 55.1% w w^−1^ for moisture, providing RMSEP of 2.1 and 0.9%, respectively. The performance of near- and mid-infrared spectroscopy to determine several quality parameters of cheese, including moisture and fat was compared [41] and the results showed that near-infrared spectroscopy was more accurate than mid-infrared, with RPD of 2.14 and 3.38% for fat and moisture, respectively [41]. These infrared procedures showed better predictability than the results acquired with TD-NMR. However, the proposed TD-NMR relaxometry method is interesting because it is applicable in through-package determinations, which is not possible with the infrared ones. Thus, the RC parameters can be predicted using TD-NMR relaxometry and chemometrics without violating the seal of packages.

## 4. Conclusions

Wide bore low field time domain NMR (TD-NMR) relaxometry data, modeled by chemometrics methods, can be a fast and non-invasive procedure to predict fat, defatted dry mass, and moisture content in RC samples. The models were developed using 45 commercially available RC samples (27 traditional, 9 light, 4 vegan, 3 lactose-free, and 2 fibers). The PLS models obtained with T_1_ relaxation time measured with a single shot CWFP-T_1_ pulse showed better predictability than the models evaluated with T_2_ relaxation time measured with the CPMG pulse sequence. In addition, the PLS model performances were improved when combined with OPS variable selection method. Therefore, CWFP-T_1_ data modeled with chemometrics can be a fast method to monitor the quality of RC directly in commercial packages. However, the potential of CWFP-T_1_ sequence in TD-NMR spectrometer still needs to be verified with a larger number of RC samples to obtain a robust regression model to predict fat, moisture, and defatted dry matter.

## Figures and Tables

**Figure 1 molecules-27-04434-f001:**
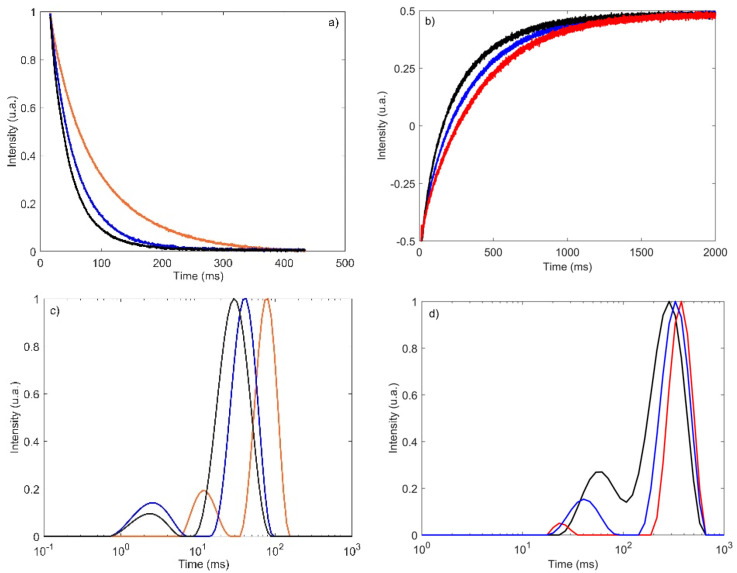
(**a**) CPMG and (**b**) CWFP-T_1_ relaxation curves of three RC samples with different chemical compositions and the respective relaxation spectra (**c**,**d**) obtained with the ILT algorithm [35]. RC samples with the lowest, medium, and the highest moisture contents are the black, blue, and red lines, respectively.

**Figure 2 molecules-27-04434-f002:**
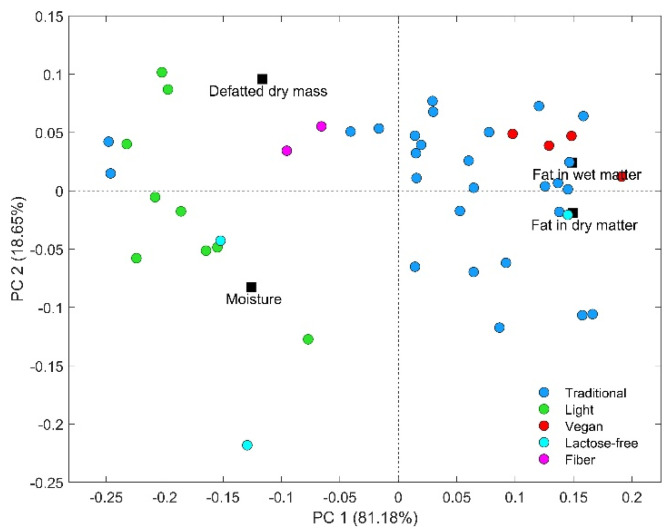
Scores (circles) and loadings (squares) biplot of the first two components of a PCA model of the RC samples.

**Figure 3 molecules-27-04434-f003:**
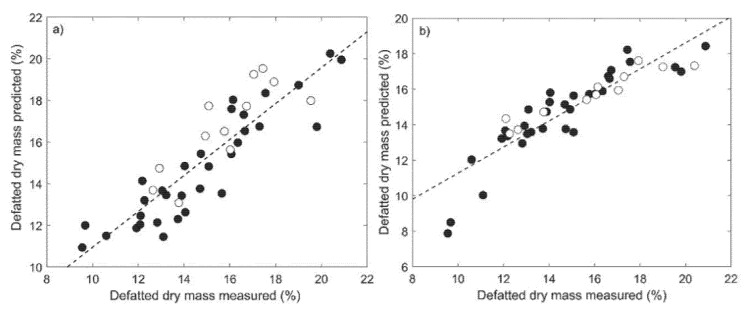
Plots of reference versus predicted values for the calibration (black circles) and validation (white circles) samples for (**a**) CMPG and (**b**) CWFP-T_1_.

**Figure 4 molecules-27-04434-f004:**
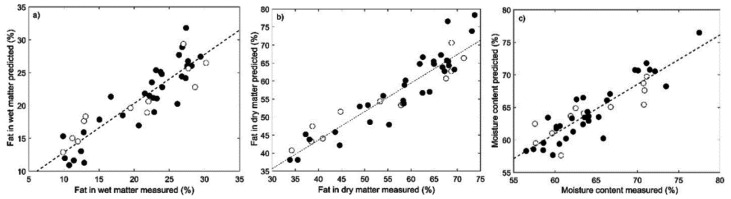
Plots of the reference versus predicted values for the calibration (black circles) and validation (white circles) samples for (**a**) FWM, (**b**) FDM, and (**c**) moisture obtained with CWFP-T₁ data.

**Table 1 molecules-27-04434-t001:** Performance parameters of the PLS models obtained using all variables and those selected by the OPS algorithm for DDM quantification.

	CPMG		CWFP	
	Full	OPS	Full	OPS
**Nvars ***	993	205	5965	75
**LV**	3	3	3	3
**RMSEP ****	1.51	1.53	1.84	1.38
**R^2^_pred_**	0.49	0.67	0.82	0.90
**RPD_pred_**	1.37	1.36	1.46	1.95
**REP_pred_ ****	9.55	9.67	11.60	8.70

* number of variables. ** % w w^−1^.

**Table 2 molecules-27-04434-t002:** PLS models obtained for FDM, FWM, and moisture quantification using the OPS algorithm.

	CPMG			CWFP-T₁		
	Fat in Dry Matter	Fat in Wet Matter	Moisture	Fat in Dry Matter	Fat in Wet Matter	Moisture
**RMSEP ***	10.90	6.12	4.97	4.71	3.28	3.00
**R^2^_pred_**	0.28	0.23	0.050	0.92	0.85	0.70
**RPD_pred_**	1.26	1.24	1.07	2.92	2.32	1.77
**REP_pred_ ***	20.35	31.13	7.71	8.79	16.68	4.65

* % w w^−1^.

## Data Availability

The data presented in this study are available on request from the corresponding author.

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
