# Peer review of "Non-Invasive Method to Predict the Composition of Requeijão Cremoso Directly in Commercial Packages Using Time Domain NMR Relaxometry and Chemometrics"

_molecules, 2022, doi:10.3390/molecules27144434_

Round 1

Reviewer 1 Report

The manuscript by Machado and coworkers provides a nice demonstration of the use of low-field NMR along with PCA analysis to provide quality parameters (fat, moisture, dry matter) of cheese in original packing. The manuscript is well prepared and the chosen methods relevant for the study. The manuscript is may be published upon conderation of the following minor points:

1) T2 and T1 data is compared and it is judged that the T1 data provide basis for the most accurate results, although both seems useful using two (fast and slow) relaxation components. From ILT it appears that the dominant signal for T2 is varying substantially in relaxation time, while the minor signal is less dependent on the matrix. The opposite applies to the T1 measurements. It would be relevant to discuss these components in more detail and most important include also a PCA analysis where both T1 and T2 information is included.

2) PCA analysis of low-field NMR relaxation data has proven useful to establish functionality between measured and known parameters for calibration sets to be validated in additional measurements. The number of data is not very large and it would be good to either include more data or at least discuss in more detail accuracies of the established method without using an excessive number of digits.

3) It may relevant to discuss the application step for industrial/comercial use. Can the results (experiments, PCA models, etc) be used directly or will further application require repetition of the study to reestablish model and parameters+ 

4) 17 of the 41 cited papers involves the senior author of this paper, which may not adequately reflect the field. It is relevant to balance referencing more appropriately 

Author Response

Reviewer 1

The manuscript by Machado and coworkers provides a nice demonstration of the use of low-field NMR along with PCA analysis to provide quality parameters (fat, moisture, dry matter) of cheese in original packing. The manuscript is well prepared and the chosen methods relevant for the study. The manuscript is may be published upon conderation of the following minor points:

1) T2 and T1 data is compared and it is judged that the T1 data provide basis for the most accurate results, although both seems useful using two (fast and slow) relaxation components. From ILT it appears that the dominant signal for T2 is varying substantially in relaxation time, while the minor signal is less dependent on the matrix. The opposite applies to the T1 measurements. It would be relevant to discuss these components in more detail and most important include also a PCA analysis where both T1 and T2 information is included.

Response: A PCA analysis was evaluated as suggested by the referee and the result obtained was exactly the same as those shown in Figure 2, indicating that the T1 e T2 values do not contribute to the samples to the separation. For this reason, this result was not included in the revised manuscript. We add the text “data not shown” in the revised manuscript.

2) PCA analysis of low-field NMR relaxation data has proven useful to establish functionality between measured and known parameters for calibration sets to be validated in additional measurements. The number of data is not very large and it would be good to either include more data or at least discuss in more detail accuracies of the established method without using an excessive number of digits.

Response: We agree that this is an important point in many multivariable studies, however there are a limited total number of types of real commercial samples available in the market for our evaluation, and we included all we could. More details of the RPD validity were included in the revised manuscript. The number of digits in the % of PCA analysis was re-written in the manuscript.

3) It may relevant to discuss the application step for industrial/comercial use. Can the results (experiments, PCA models, etc) be used directly or will further application require repetition of the study to reestablish model and parameters+ 

Response: As we the model was constructed with patronized commercial samples, for the used spectrometer/magnets there are no need for further development of the method, however, in different TD-NMR spectrometer/magnets such model needs to be reestablish.

4) 17 of the 41 cited papers involves the senior author of this paper, which may not adequately reflect the field. It is relevant to balance referencing more appropriately 

Response: We accepted the suggestion, and some cited papers were included and changed. Two cited papers of senior researcher were deleted

Reviewer 2 Report

Although the possibility to predict the composition of the “requeijao cremoso” cheese directly in the package by NMR relaxometry might be of interest, the proposed article lacks an adequate experimental design.

Major issues:

11.       Concerning the application of multivariate analysis:

a.       it appears that the authors have applied PCA and PLS multivariate analysis on the collection of the whole decays from CPMG and/or CWFP-T1. The goal of PCA and PLS is to highlight latent variables, which are linear combinations of the original variables, within a data set.  But the decays are ruled by an exponential function (biexponentiality was found by the authors in this case). Therefore there is no point in applying PCA and PLS in this case: for instance, the “almost linear” parts of the decays (the steep first part and the “plateau” last part) which are similar in any decay (see fig. 1) will provide strong correlations;

b.       Multivariate methods like PCA and PLS are based on the assumption that the number of variables is smaller than the number of samples (ideally, much smaller). If this is not the case, like in the present article, the results are not reliable even if all the internal quality parameters seem to support the goodness of the achieved models;

The authors might replace the “full TD-NMR decay” in the analysis with the T1,x and T2,x values;

c.       The sample subpopulations are strongly unbalanced. Over 45 samples, 27 are “traditional” RC, the remaining subpopulations are composed by 9, 4, 3 and 2 samples, respectively. As a consequence, the results of the PCA analysis is not representative because the correlation matrix is dominated by the correlations between the variables of the “traditional” RC samples. This makes the multivariate analysis highly questionable, as well as any following consideration. In the present form, no conclusion can be drawn about differences among the RC types;

22.       NMR analysis: since the length of the 90° pulse is 33 ms, the authors should explain why the 180° pulse is 60 ms and the 18° one is 4m ms;

33.       The limits set for RPD validity should be supported more strongly: the sentences in lines 265-267 are copied from ref. 39, which in turn does not give explanations but refers to Patil (2010) and  Williams P (2001). On his side Patil refers to Williams P., therefore the only available source is Williams P (2001) “Implementation of near-infrared technology”. In: Williams P, Norris K (eds) Near-infrared technology in the agricultural and food industries, 2nd edn. American Association of Cereal Chemists Inc., St. Paul, which is not easily available;

44.       Most of the references regarding the variable reduction are articles in which IR spectroscopy is used: the authors should provide references supporting the use of such reduction technique in NMR relaxometry.

Minor issues:

11.       some reference in the text should be replaced by their numbers (lines 55, 58, 308, 312);

22.       line 110: “with approximately 200 g” not clear;

33.       line 143: the link brings to a web page in Portuguese, where it is not obvious from where the Matlab routine can be downloaded;

44.       line 162: does figure 1a display all of the TD-NMR signals, or just some of them as stated for the CWFP-T1 in figure 1b?

55.       A legend with symbols and colors would make figure 2 easier to read, rather than describing symbols and colors in the caption.

Author Response

Reviewer 2

Although the possibility to predict the composition of the “requeijao cremoso” cheese directly in the package by NMR relaxometry might be of interest, the proposed article lacks an adequate experimental design.

Major issues:

  1. Concerning the application of multivariate analysis:
  2. it appears that the authors have applied PCA and PLS multivariate analysis on the collection of the whole decays from CPMG and/or CWFP-T1. The goal of PCA and PLS is to highlight latent variables, which are linear combinations of the original variables, within a data set.  But the decays are ruled by an exponential function (biexponentiality was found by the authors in this case). Therefore there is no point in applying PCA and PLS in this case: for instance, the “almost linear” parts of the decays (the steep first part and the “plateau” last part) which are similar in any decay (see fig. 1) will provide strong correlations;

Response: Our system is not composed by two singular exponential times (pure bi-exponential). The real system is complex, been much better represented by two large distribution of relaxation times in the ILT spectra. We understand that the multivariate methods will express the differences in these signals, resulting that the “plateau” parts of the signal do not contribute to the group separations. This is a usual approach in the multivariable literature for TD-NMR, high-resolution NMR spectra, and other spectroscopies.

  1. Multivariate methods like PCA and PLS are based on the assumption that the number of variables is smaller than the number of samples (ideally, much smaller). If this is not the case, like in the present article, the results are not reliable even if all the internal quality parameters seem to support the goodness of the achieved models;

Response: In contrast to univariate calibration, which works with a single instrumental response measured for each experimental sample, multivariate calibration works with many different signals for each sample. Depending on the instrumental setup, the delivered data for a single sample may have different degrees of complexity. The simplest multivariate data are those produced in vector form, which is explored in this study. The literature on this field is abundant and its advantages compared with univariate methods can be found in the study published by Ramus Bro (Analytica Chimica Acta 500 (2003) 185–194).

The authors might replace the “full TD-NMR decay” in the analysis with the T1,x and T2,x values;

Response: We prefer to use the “full TD-NMR decay” because it contains more information of the signal and the T1 and T2 values, determined by fitting methods limits the information only to two or three components and not to the full information observed in the signals. In this sense, we believe that the approaches using the overall data are more reliable than using a preprocessing method in the data, to transform it in two singular exponential values.

  1. The sample subpopulations are strongly unbalanced. Over 45 samples, 27 are “traditional” RC, the remaining subpopulations are composed by 9, 4, 3 and 2 samples, respectively. As a consequence, the results of the PCA analysis is not representative because the correlation matrix is dominated by the correlations between the variables of the “traditional” RC samples. This makes the multivariate analysis highly questionable, as well as any following consideration. In the present form, no conclusion can be drawn about differences among the RC types;

Response: We agree that this is a good critical point in many multivariable studies, however there are a limited total number of types of real commercial samples available in the market for our evaluation, and we included all we could acquire.

  1. NMR analysis: since the length of the 90° pulse is 33 ms, the authors should explain why the 180° pulse is 60 ms and the 18° one is 4m ms;

Response: Our NMR probe has a bore of 10 cm, and the calibration of the pulses show that 90° pulse is 33 ms and 180° pulse is 60 ms. For the 4 ms pulse, it was wrong written that represents a 18°, where the corrected value is 10°, which we corrected in the revised text.

  1. The limits set for RPD validity should be supported more strongly: the sentences in lines 265-267 are copied from ref. 39, which in turn does not give explanations but refers to Patil (2010) and  Williams P (2001). On his side Patil refers to Williams P., therefore the only available source is Williams P (2001) “Implementation of near-infrared technology”. In: Williams P, Norris K (eds) Near-infrared technology in the agricultural and food industries, 2nd edn. American Association of Cereal Chemists Inc., St. Paul, which is not easily available;

Response: More references were included in the revised manuscript in order to support the limits set of RPD used in this study.

  1. Most of the references regarding the variable reduction are articles in which IR spectroscopy is used: the authors should provide references supporting the use of such reduction technique in NMR relaxometry.

Response: One important aspect explored in this study is the use of variable selection for improving the analytical performance of the developed methods. To the best of our knowledge, it is the first time that the variable selection method is applied to TD-NMR data. However, variable selection methods have been in use for at least 50 years aiming to improve the performance of the models by removing many irrelevant, noisy or unreliable variables.

Minor issues:

  1. some reference in the text should be replaced by their numbers (lines 55, 58, 308, 312);

Response: Sorry by this mistake that was corrected in the revised manuscript.

  1. line 110: “with approximately 200 g” not clear;

Response: The sentence was rewritten to become clearer.

  1. line 143: the link brings to a web page in Portuguese, where it is not obvious from where the Matlab routine can be downloaded;

Response:  A relevant reference to the ordered predictors selection (OPS) method was included. Furthermore, although the web page cited is in Portuguese, the download section is in English.

  1. line 162: does figure 1a display all of the TD-NMR signals, or just some of them as stated for the CWFP-T1in figure 1b?

Response: Figure 1 shows the TD-NMR signals of three RC samples with different chemical compositions acquired using CPMG (Fig. 1a) and CWFP-T₁ (Fig. 1b) pulse sequences.

  1. A legend with symbols and colors would make figure 2 easier to read, rather than describing symbols and colors in the caption.

Response: We include a legend in figure 2 as suggested.

Round 2

Reviewer 2 Report

Application of multivariate analysis to the whole decays:

The authors cited the paper by Rasmus Bro, where more suitable methods like multi-way ones are described to be better suited to analyze data like those get by the authors. PCA and PLS are used for noise reduction and outliers identification, can be further used only for exploratory purposes and are not ideal to achieve their goals, weakening the strength of their findings. This is true also for their choice to use the whole decays instead of the T1 and T2 values. In my opinion these are very questionable choices, but this discussion does not imply that their point of view must be rejected.

Experimental design:

The fact that there is a limited number of samples available is not a valid reason to apply a strongly unbalanced samples population. If the authors want to use such a dataset, they must state this pitfall very clearly within the text ("results and discussion" and "conclusion" sections), together with the caveat that the correlation matrix is dominated by the “traditional” RC samples and is therefore not representative for all the samples. A more elegant approach might be performing a PLS model based only on “traditional” RC data and check whether the datasets of the remaining groups are recognized by that model (e.g. by PLS, or SIMCA) drawing the due conclusions.

Editing:

The "Conclusion" section should be numbered as 4 and not 5.

Author Response

Reviewer 2:

Application of multivariate analysis to the whole decays:

The authors cited the paper by Rasmus Bro, where more suitable methods like multi-way ones are described to be better suited to analyze data like those get by the authors. PCA and PLS are used for noise reduction and outliers identification, can be further used only for exploratory purposes and are not ideal to achieve their goals, weakening the strength of their findings. This is true also for their choice to use the whole decays instead of the T1 and T2 values. In my opinion these are very questionable choices, but this discussion does not imply that their point of view must be rejected.

Response: We thank the referee for all comments and careful evaluation of the manuscript. In this study, PCA was applied for exploratory purposes in order to observe similarities and/or differences between the different types of RC samples. Conversely, the correlations between chemical (moisture, FDM, FWM, and DDM) and TD-NMR data were evaluated using partial least squares (PLS) regression. PLS is a common method applied in spectroscopy to maximize the correlation between spectral data and the parameters to be quantified. The use of the whole TD-NMR decays for the development of the PLS models was based on previous studies reported in the literature (1-5). In these studies, regression models were evaluated using different approaches (univariate and multivariate analysis), and the results obtained applying the whole TD-NMR decay give good results with a high correlation and low error.

In our study, regression models were also evaluated by correlating the T1 or T2 values against the moisture, fat, and defatted dry matter contents (univariate approaches) and combining T1 and T2 values (multivariate approaches). However, results obtained with the full TD-NMR decays showed better predictability, with higher correlation coefficient. Therefore, only these results were shown in detail in the manuscript. This information was included in the revised manuscript (lines 300 to 304).

  1. Pedersen, H.T., Munck, L. & Engelsen, S.B. (2000). Low-field 1H nuclear magnetic resonance and chemometrics combined for simultaneous determination of water, oil, and protein contents in oilseeds. J Amer Oil Chem Soc 77, 1069–1077.
  2. Jepsen, S. M., Pedersen, H. T., & Engelsen, S. B. (1999). Application of chemometrics to low-field 1H NMR relaxation data of intact fish flesh. Journal of the Science of Food and Agriculture, 79, 1793–1802.
  3. Hansen, C. L., Thybo, A. K., Bertram, H. C., Viereck, N., Van Dan Berg, F., Engelsen, S. B. (2010). Determination of dry matter content in potato tubers by Low-Field Nuclear Magnetic Resonance (LF-NMR). J. Agric. Food Chem., 58, 10300–10304.
  4. Ramos, P.F.O., Toledo, I.B., Nogueira, C.M., Novotny, E.H., Vieira, A.J.M., Azeredo, R.B.V. (2009). Low field 1H NMR relaxometry and multivariate data analysis in crude oil viscosity prediction. Chemometr Intell Lab Syst., 99, 21–126.
  5. Santos, P. M., Pereira-Filho, E. R., & Colnago, L. A. (2016). Detection and quantification of milk adulteration using time domain nuclear magnetic resonance (TD-NMR). Microchemical Journal, 124, 15–19.

Experimental design:

The fact that there is a limited number of samples available is not a valid reason to apply a strongly unbalanced samples population. If the authors want to use such a dataset, they must state this pitfall very clearly within the text ("results and discussion" and "conclusion" sections), together with the caveat that the correlation matrix is dominated by the “traditional” RC samples and is therefore not representative for all the samples. A more elegant approach might be performing a PLS model based only on “traditional” RC data and check whether the datasets of the remaining groups are recognized by that model (e.g. by PLS, or SIMCA) drawing the due conclusions.

Response: As suggested by the reviewer, the information about the sample subpopulations was included in the "results and discussion" and "conclusion" sections of the revised manuscript (lines 194-195 and lines 327-327).

Besides, PLS models were evaluated according to the referee’s suggestion and the results obtained were similar to those shown in the manuscript. PLS-DA and SIMCA models were evaluated to assign the RC samples, however, a poor performance was obtained.

In our study, Kennard-Stone algorithm was applied to split the sample into calibration and validation sets. This procedure assures the selection of representative and homogeneously distributed samples in the whole analytical range of the multivariate space.

Editing:

The "Conclusion" section should be numbered as 4 and not 5.

Response: We apologize for that. We have changed in the revised manuscript.
